



# Effects of surface current/wind interaction in an eddy-rich general ocean circulation simulation of the Baltic Sea

H. Dietze[1] and U. Löptien[1]

[1]GEOMAR, Helmholtz Centre for Oceanic Research Kiel, Düsternbrooker Weg 20, 24105 Kiel, Germany

*Correspondence to:* H. Dietze
(hdietze@geomar.de)

**Abstract.** Deoxygenation in the Baltic Sea endangers fish yields and favours noxious algal blooms. Yet, vertical transport processes ventilating the oxygen-deprived waters at depth and replenishing nutrient-deprived surface waters (thereby fuelling export of organic matter to depth), are not comprehensively understood. Here, we investigate the effects of the interaction between surface currents and winds (also referred to as eddy/wind effects) on upwelling in an eddy-rich general ocean circulation model of the Baltic Sea. Contrary to expectations we find that accounting for current/wind effects does inhibit the overall vertical exchange between oxygenated surface waters and oxygen-deprived water at depth. At major upwelling sites, however, as e.g. off the south coast of Sweden and Finland, the reverse holds: the interaction between topographically steered surface currents with winds blowing over the sea results in a climatological sea surface temperature cooling of $0.5\,\mathrm{K}$. This implies that current/wind effects drive substantial local upwelling of cold and nutrient-replete waters.



# 1 Introduction

Almost a century ago, Taylor et al. (1916) proposed that the stress describing the exchange of momentum between a moving atmosphere and the earth's surface may be expressed as being proportional to the windspeed squared times the density of air. As for the respective proportionality constant, often referred to as the drag coefficient, Taylor et al. (1916) calculated values
between 0.002 and 0.003 " ... for the ground at Salisbury Plain, where the wind observations were made".

Numerous studies, targeted at improving the fidelity of this seminal relationship for oceanic applications, have been published ever since. Among them those exploring the dependency of the drag coefficient on (a) wind speed (e.g. Smith and Banke, 1975), (b) atmospheric stability (e.g. Hsu, 1974) and (c) wind-wave interaction (e.g. Hsu, 1973). Even so, the transfer of momentum from the atmosphere to the ocean is still associated with considerable uncertainties. For example, data assimilation
experiments suggest that recent surface stress estimates have to be substantially corrected in order to reconcile *in situ* observations with circulation models (Stammer et al., 2004). To this end, it is somewhat conspicuous, that the respective wind stress corrections are especially large in regions of strong surface currents such as in the Gulf Stream, Kuroshio, Leeuwin Current and the Antarctic Circumpolar Current. The nearby conclusion, that the drag coefficient is substantially influenced by strong ocean currents is, however, supposedly wrong (Kara et al., 2007).

In contrast to ongoing controversial discussions on the drag coefficient, there is now consensus that the calculation of the stress, exerted by the wind on a circulating ocean, should be based on the wind vectors relative to the ocean currents - rather than being based on the wind vectors only (Fairall et al., 2003). This may be surprising given the rather high uncertainties of the drag coefficient (discussed above), and that ocean currents are typically orders of magnitudes smaller than atmospheric winds which – in turn – suggests that the neglect of the (slow) movement of the ocean's surface should not alter the stress
calculation significantly within its already-substantial uncertainty. Martin and Richards (2001) put forward a striking argument to consider the ocean's movement at the surface in the stress calculation nonetheless. It is based on the success of the concept *Ekman Pumping*. *Ekman Pumping* is considered a major agent via which the atmosphere drives the general oceanic circulation (Stommel, 1957) and it is key to our understanding how energy is transferred into the interior ocean (e.g. Roquet et al., 2011). *Ekman Pumping* depends on the curl of the wind stress which is composed of spatial derivatives; and thus comes the relevance
of ocean surface currents: oceanic currents typically vary on spatial scales that are much smaller than the winds associated to relatively large-scale atmospheric weather systems. So the argument here is that, in terms of their effect on *Ekman Pumping* which is a major control on circulation, ocean surface currents compensate for their rather weak magnitude by relatively large changes on small spatial scales.

The study of Martin and Richards (2001) includes a drastic demonstration of this effect. The authors argue that even a
spatially uniform wind blowing over an ocean eddy should yield significant wind stress curls and an associated vertical *Ekman Pumping* of the order of $\approx 0.5\,\mathrm{m\,day^{-1}}$ in typical open-ocean conditions. Direct observations of this so-dubbed *eddy/wind effect* in a North Atlantic eddy by McGillicuddy et al. (2007) and Ledwell et al. (2008) confirmed both the existence of the process and its magnitude. The associated local effects on marine biota were so pronounced that the authors speculated that the, hitherto unaccounted, eddy/wind effect could resolve a long-standing discrepancy between nutrient supply to, and oxygen





consumption below the euphotic zone of the subtropical gyre (e.g. Oschlies et al., 2003; Dietze and Oschlies, 2005; Kähler et al., 2010). This, however, has been discussed controversially by Eden and Dietze (2009).

This study sets out to constrain the effects of surface current/wind interaction in the Baltic Sea, a marginal sea in central northern Europe where eddy/wind effects should be especially prominent: as explained above, the spacial scales of surface currents are key to the magnitude of the effect. Rough scaling suggests that an energetic prevalent spacial scale of oceanic circulation is the Rossby radius of deformation. In typical mid-latitude open-ocean conditions (such as explored by Eden and Dietze, 2009) the Rossby radius is of the order of $50\,\mathrm{km}$ and the associated eddy/wind effect would be of the order of $\approx 0.5\,\mathrm{m\,day^{-1}}$, as explained above. In the Baltic Sea, however, effected by shallower water depths and strong stratification, the Rossby radii are typically an order of magnitude smaller (1.3 to $7\,\mathrm{km}$, Fennel et al., 1991) which, in turn, suggests that eddy/wind effects are an order of magnitude stronger than in the open ocean. If so, this effect should be a major process in the Baltic effecting vertical exchange between the well-oxygenated surface waters and the dense oxygen-depleted deep waters. As such surface current/wind effects should rank among the first oder processes controlling the intermittent deoxygenation of the Baltic Sea which endangers fish yields and favours noxious algal blooms.

So far, surface current/wind effects have not been explicitly investigated in the Baltic. Among the reasons are the small Rossby radii in the Baltic which call for much higher horizontal resolution and associated computational cost than is the case for open-ocean model studies. It is only recently that advances in compute hardware have rendered this feasible.

This paper sets out to explore current/wind effects in the Baltic with the recently-developed high-resolution general ocean circulation model configuration MOMBA 1.1 (Dietze et al., 2014). Section 2 describes the model configuration and the numerical experiments. Section 3 presents model results followed by a discussion in Section 4. We close with a summary and conclusions in Section 5.

## 2 Method

We conduct a numerical twin experiment with the general circulation ocean model configuration MOMBA 1.1 (Dietze et al., 2014): the reference simulation *REF* includes the surface current/wind effect while the other simulation *noCW* (short for **no c**urrent/**w**ind effect) does not account for this effect. A major difference compared to previous studies that cover the Western Mediterranean (Olita, 2015) and the North Atlantic (Eden and Dietze, 2009), is our focus on the shallow Baltic Sea where, as outlined above, current/wind effects should be most prominent.

### 2.1 Model Configuration

All experiments are based on a regional ocean-ice model setup of the Baltic Sea, called MOMBA 1.1. The configuration is extensively documented in Dietze et al. (2014), and accessible via www.baltic-ocean.org. The model features a competitive horizontal resolution of $\approx 1$ nautical mile, it is eddy-rich in that it starts to resolve the relevant spacial scales (c.f. Fennel et al., 1991). The model domain is bounded by $4.2°\mathrm{E}$ and $30.3°\mathrm{E}$ and $53.8°\mathrm{N}$ to $66°\mathrm{N}$. The vertical discretisation comprises 47 levels. There are no open boundaries, i.e., the model domain is surrounded by solid walls. We use the KPP boundary layer





parameterisation (Large et al., 1994) with parameters identical to those applied in the eddy-permitting global configurations of Dietze and Kriest (2012); Dietze and Löptien (2013) and Liu et al (2010). The atmospheric boundary conditions driving MOMBA 1.1 are based on dynamically downscaled ERA-40 reanalysis data (Uppala et al., 2005). The respective downscaling is performed by the Rossby Centre Regional Atmosphere model version 3 (hereafter RCA3), which takes ERA-40 reanalysis

data as boundary conditions. RCA3 features an enhanced (relative to ERA-40) horizontal resolution of 25 km (Jones et al., 2004; Samuelsson et al., 2011). These atmospheric boundary conditions are identical to those applied to the "RCO"-models, used e.g. by Hordoir et al. (2013); Löptien et al (2013) and Meier and Faxen (2002).

Results from a 1987 to 1999 hind cast simulation showcased in Dietze et al. (2014) illustrate that the model's fidelity is competitive to other Baltic Sea models. Remarkably, MOMBA 1.1 simulates very realistic sea surface temperatures which is

indicative of a correct representation of near-surface diabatic processes.

## 2.2   Experiments

In the following Section 3, we compare two numerical experiments, *REF* and *noCW*. *REF* refers to the reference simulation including the surface current/wind effect, while the other simulation *noCW* does not account for the current/wind effect. Note that *REF* is identical to MOMBA 1.1 and has, in contrast to earlier Baltic Sea models, a better representation of the feedback

of ocean surface currents, $\boldsymbol{u}_o$, on the wind stress, $\boldsymbol{\tau}_{ref}$ because (a) it explicitly accounts for ocean currents in the calculation of the wind stress as recommended by e.g., Large and Yeager (2004); Fairall et al. (2003), and (b) in contrast to the previous generation of ocean circulation models, MOMBA 1.1 features a relatively high horizontal resolution of 1 nautical mile which starts to resolve mesoscale processes in the Baltic.

The stress $\boldsymbol{\tau}_{ref}$ exerted on the ocean's surface by winds blowing with a velocity $\boldsymbol{u}_a$ over an oceanic current moving with a

velocity $\boldsymbol{u}_o$ is calculated in experiment *REF* as:

$$\boldsymbol{\tau_{ref}} = \rho_a c_D |\boldsymbol{u}_a - \boldsymbol{u}_o| (\boldsymbol{u}_a - \boldsymbol{u}_o), \tag{1}$$

with the density of air $\rho_a$, and the dimensionless drag coefficient $c_D$. Note that $c_D$ is not constant as we apply the formulation detailed in Large and Yeager (2004); Large (2006) which has matured to a reference in the field (e.g. Griffies et al., 2014).

The setup *noCW* is identical to *REF* except for that the traditional (similar to, e.g., Meier et al., 1999, their Eq. 30), physically

less plausible way to force an ocean model, which neglects the effect of surface currents on the wind stress, is applied. More specifically, the stress $\boldsymbol{\tau}_{noCW}$ exerted on the ocean's surface is calculated in experiment *noCW* as:

$$\boldsymbol{\tau_{noCW}} = \rho_a c_D |\boldsymbol{u}_a| \boldsymbol{u}_a. \tag{2}$$

Both simulations start from rest on 1 January 1987 and are integrated till 31 December 1993. In the following analysis we explore the model output starting from 1 January 1988 thus allowing for a 1 year spin-up phase. We stop the simulation on

31 December 1993 which constrains the period to an apparently especially realistic model behaviour. Contrary to Dietze et al. (2014) we apply compiler options which ensure bit-reproducibility (c.f. Sec. 3.8 in Dietze et al., 2014).





## 3   Results

One major focus here is on simulated sea surface temperatures (SSTs) and causes of SST differences in the simulations *REF* and *noCW*. We argue that SST is prominent because (1) it controls the velocity of biogeochemical turnover in the sun-lit surface, as e.g. enzyme-catalyzed reactions feature a sensitivity corresponding to roughly 10% increase per Kelvin increase. (2) SST controls sea fog. This is of interest since the Baltic hosts up to 15% of the world's international maritime cargo (HELCOM , 2009) and because around 10% of all collisions are apparently related to sea fog (Tuovinen et al., 1984). (3) SST variation is a proxy for diabatic processes. Consequently, simulated SST differences can be related to differing nutrient transports to the sun-lit surface, associated export of organic matter to depth and oxygen consumption at depth.

### 3.1   Basin-scale effects

Fig. 1 shows SST differences between the simulations *REF* and *noCW* averaged over the entire model domain. On a basin-scale, we find warmer SSTs in summer and colder SSTs in winter, when accounting for surface current/wind effects. At the same time, current/wind effects dampen the amplitude of the seasonal cycle of net air-sea heat fluxes which warm the sea in summer and cool it in winter (not shown). The combination of an amplified seasonal cycle in SST and dampened seasonal air-sea heat fluxes is indicative of a reduced "thermal momentum" of the ocean's surface layer. From this, we conclude that diabatic transport of heat is reduced by current/wind effects such that less of the water column is exposed to atmospheric heat fluxes. This conclusion is contrary to expectations outlined in the introduction. We would rather have expected an increase in diabatic transports and, consequently, an increase of the "thermal momentum" of the surface layer.

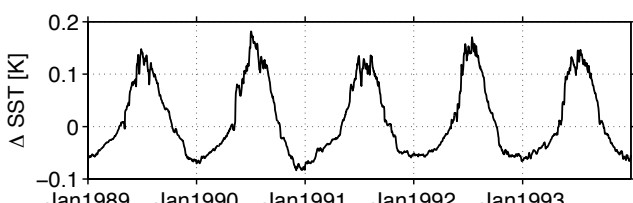

**Figure 1.** Basin-scale, sea surface temperature differences (calculated as experiment *REF - noCW*). Positive (negative) values indicate warming (cooling) by surface current/wind effects.

However, reduced diabatic transports associated with current/wind effects become reasonable when considering the transfer of kinetic energy to the ocean. A gedankenexperiment reveals that by accounting for the ocean's movement in the calculation of wind stress exerted on the ocean's surface – overall – less energy is transferred to the ocean: winds and surface currents can – in addition to having a perpendicular component to one another – either oppose one another, or run along into the same direction. Components aligned perpendicular do not yield additional (or less) stress effected by surface currents. If the winds oppose the surface currents, the ocean's movement is slowed down with the current/wind effect accelerating the oceanic energy drain. If the wind runs along with the surface current the current/wind effect reduces the drag such that less




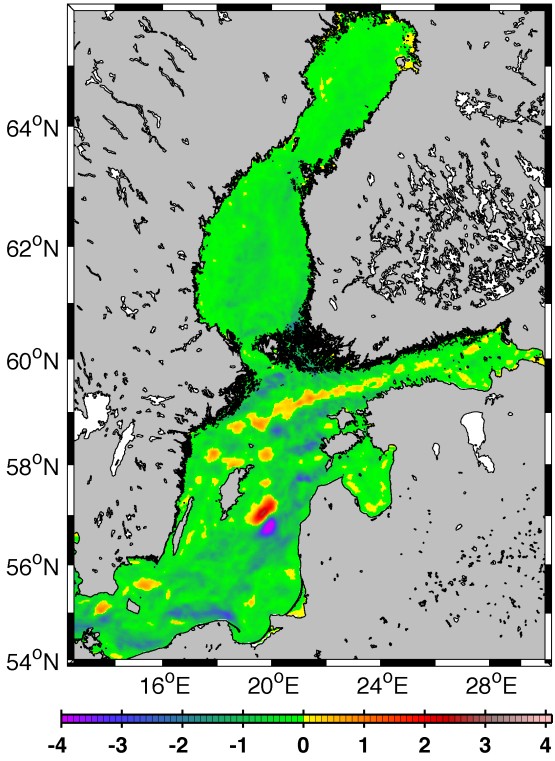

**Figure 2.** Differences in power driving the oceanic surface circulation in units $\mathrm{mW\,m^{-2}}$ (calculated as experiment *REF - noCW*, averaged over the simulated years 1989 - 1993). Positive (negative) values indicate that more (less) kinetic energy is supplied to the ocean if surface current/wind effects are accounted for.

momentum and energy is transferred to the ocean. Fig. 2 confirms this gedankenexperiment in that is shows that in experiment *REF* the net energy transfer by winds is less than in experiment *noCW*. This reduced net supply of kinetic energy yields weaker surface currents and, consequently, weaker vertical shear of horizontal velocities. This, in turn, reduces shear-induced turbulent mixing. Consistently with this argument we find, on average, shallower surface mixed layers in experiment *REF*: Fig. 3 shows the simulated differences in summer. The differences in winter have the same sign but are one order of magnitude larger (not shown). Shallower surface mixed layers result in a reduced "thermal momentum" of that surface layer that is in contact with air-sea heat fluxes. This results in the higher (lower) SSTs in summer (winter) that are associated to the current/wind effect.

## 3.2 Local effects in upwelling regions

The previous section showed that current/wind effects reduce the overall net supply of kinetic energy to the ocean and result in shallower surface mixed layers. Ultimately, this causes a response contrary to our initial expectation: on a basin-scale diabatic transport processes are reduced rather than enhanced.





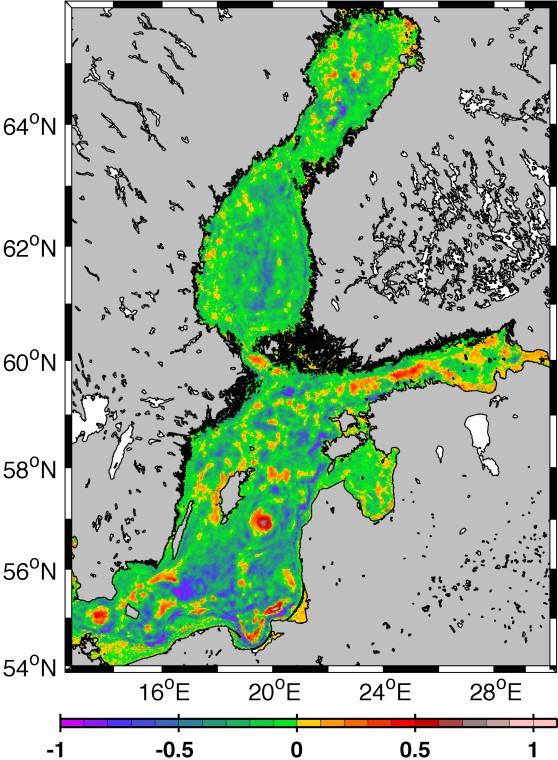

**Figure 3.** Surface mixed layer depths (as defined by Large et al., 1994, their equation 21) differences in units m (calculated as experiment *REF - noCW*, climatological July based on the simulation period 1989 - 1993). Positive (negative) values indicate a surface mixed layer that is deepened (shallowed) when surface current/wind effects are accounted for.

Fig. 4 shows that the effects are not uniform over the whole basin. Locally, and at times during the season cycle, current/wind effects do drive – consistent with initial expectations – reduced SSTs in summer. Intriguingly this applies especially to what Lehmann et al. (2012) dubbed the Baltic's "... most favourable upwelling region ..." off the southernmost coast of Sweden, off Karlshamn and off the Kalmarsund (marked by the magenta ellipse Fig. 4). An analysis of simulated local air-sea heat

5 fluxes in this region reveals that current/wind effects increase the heat supplied to the ocean by 1 to $5\,\mathrm{W\,m^{-2}}$ (compared to a basin-averaged *decrease* of $\approx 5\,\mathrm{W\,m^{-2}}$). This indicates that locally – indeed – diabatic heat fluxes must be augmented. In order to understand the underlying mechanisms we investigate winds and currents at the southernmost coast of Sweden. The winds blowing over the ocean's surface in these regions are according to Fig. 5 not peculiar: winds in July are typically northeasterly with the only irregularieties being that (1) the winds are stronger over the sea than on land (and its wake) where

10 surface roughness and associated drag is enhanced, and that (2) the winds' persistencies decreases as they travel north eastwards (not shown). What is however peculiar in the region is the distinct anticyclonic circulation in Fig. 6: the currents follow the Swedish coastline on its way to the east and return westwards some 10 nautical miles offshore. These two eastward/westward branches are rather persistent and follow closely the topography (Fig. 7). Consistently, with our initial considerations we find





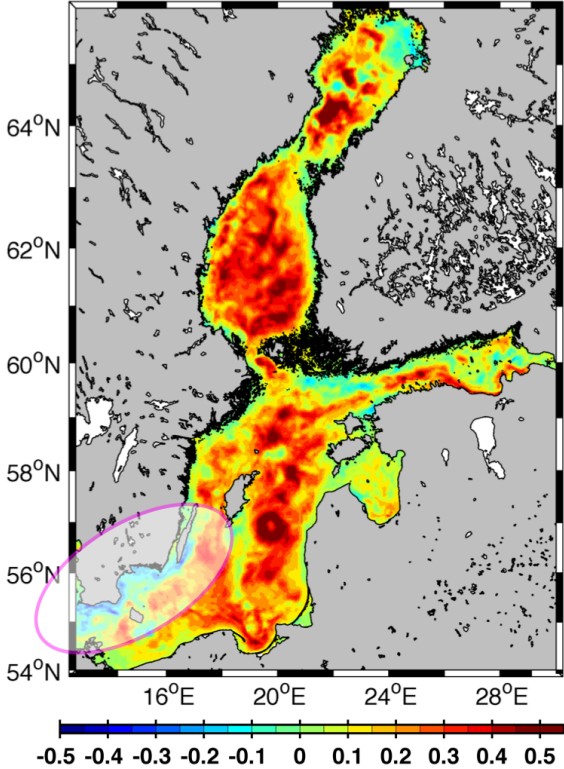

**Figure 4.** Simulated sea surface temperature differences in units K (calculated as experiment *REF - noCW*, climatological July based on simulated years 1989 - 1993). The magenta ellipse encompasses a region off southern Sweden where surface current/wind effects cool the surface.

that the winds blowing over this coastal anti-cyclonic circulation yield additional upwelling. Expressed as a climatological *Ekman pumping* representative for July the current/wind effects cause an additional local upwelling of $0.2\,\mathrm{m\,day^{-1}}$ (Fig. 8). This increase in *Ekmann Pumping* is reflected in reduced SSTs (mangenta ellipse in Fig. 4).

## 4 Discussion

5  Our numerical twin experiment yields results that are apparently inconsistent with initial theoretical considerations (Sec. 1). Based on the persuasive sketch of Martin and Richards (2001) (their Fig. 7) we expected that a proper representation of surface current/wind effects will increase diabatic transport. What we find on a basin scale, however, is the contrary as we get apparently less diapycnal mixing. In Sec. 3.1 we reconciled this inconsistency by an argumentation based on energy supply: when surface current/wind effects are accounted for, less energy is transferred to the ocean (c.f. Fig. 2).

10  An equivalent explanation can be put forward based on wind stress: the climatological, basin-scale wind stress received by the ocean is, on average, less in the configuration which accounts for surface current/wind effects than in the simulation





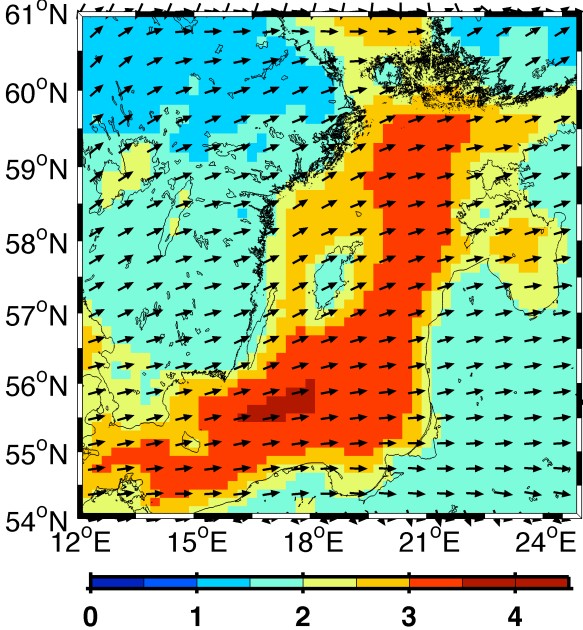

**Figure 5.** July climatology of the 10 m winds driving the simulations. The calculation is based on 1989-1993 3-hourly snapshots of RCA3 (Sec. 2.1). The arrows show the direction of the winds. The colour denotes the speed in units $\mathrm{m\,s^{-1}}$.

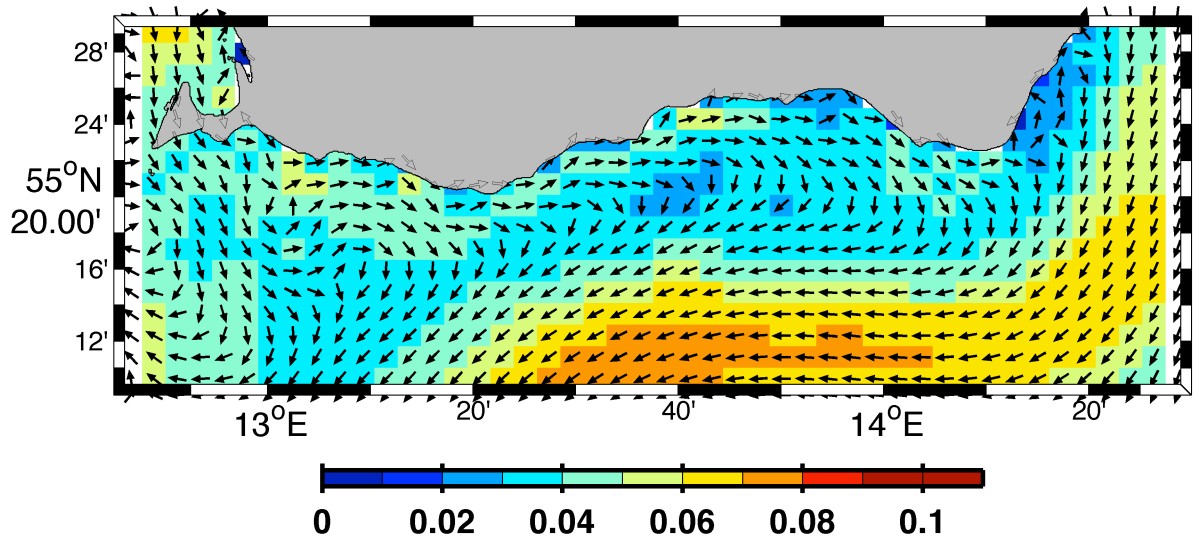

**Figure 6.** Surface circulation off the southernmost coast of Sweden in summer (climatological July calculated from model output comprising the years 1989 to 1993). The arrows show the direction of the currents. The colour denotes the speed in units $\mathrm{m\,s^{-1}}$.





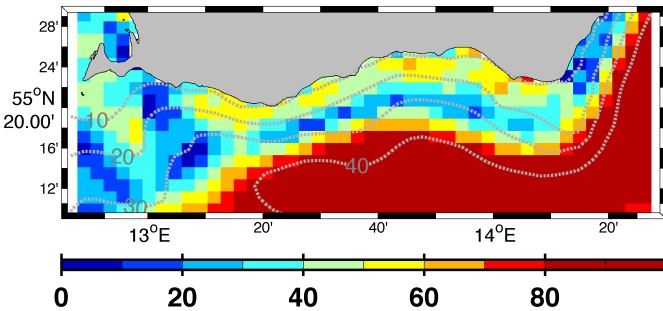

**Figure 7.** Persistency of surface currents and topography. The colour denotes the persistency of currents in climatological July in units %
(c.f. Dietze et al., 2014, their equation 15). The grey, dashed contours refer to isobaths with 10 m spacing.

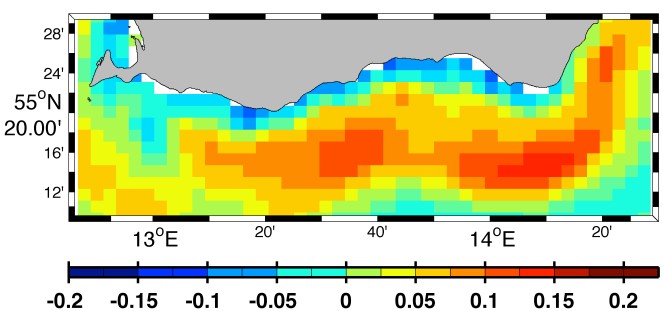

**Figure 8.** *Ekman Pumping* effected by current/wind interaction (calculated as the difference between experiment *REF - noCW*; climatological
July based on simulated years 1989 - 1993) in units $\mathrm{m\,day^{-1}}$. Positive values denote additional upwelling.

that does not account for this effect (not shown). Here, again, the explanation is associated with the fact that, (even) on a
rotating planet, the oceanographic response to wind forcing is typically enhanced surface flow in the direction of the wind (in
addition to a perpendicular component). If accounted for in the calculation of wind stress this reduces the stress acting on the
ocean. Somewhat unexpected, as regards their effect on wind stress curl and associated *Ekman Pumping*, the stress reduction
5   prevails the effects of increased horizontal inhomogeneity. This however holds only for averages in time and space. Locally,
and at times, the increased horizontal stress inhomogeneity can drive additional *Ekman Pumping*. We calculated that maximum
climatological values peak at $0.2\ m\ day^{-1}$ on the south coast of Sweden. This drives, as initially expected, indeed additional
diabatic fluxes, as is indicated by pronounced SST anomalies (particularly in summer). It is noteworthy, however, that the
magnitude of actual vertical velocities calculated from daily averages are unexpectedly small compared to *Ekman* velocities
10   diagnosed from the wind stress. This mismatch between *Ekman Pumping* calculated from the wind stress and actual vertical
velocities suggests that some preconditions mandatory for the applicability of the *Ekman Theorie* are violated. Among these
preconditions that are not met are (1) the assumption that boundaries have no effect, i.e. *Ekman Theorie* applies only to a
waterbody of infinite extend in all three spacial dimension – a certainly overoptimistic assumption in the shallow marginal



Baltic Sea. (2) The theory assumes a viscosity that is constant with depth – an assumption certainly violated given that the KPP boundary layer parameterisation (Large et al., 1994) which is applied in our simulations has been specifically designed to reproduce observed *non*-constant vertical profiles of diffusivities and viscosities.

Our model results show that accounting for current/wind effects does not drive any additional near-surface diapycnal transport on a basin-scale. This does not imply that current/wind effects are irrelevant. It merely means that the physically less plausible formulation of not accounting for them does not necessarily underestimate diapycnal fluxes because antagonistic effects are at play: on the one hand surface current/wind effects reduce the kinetic energy supply to the ocean, and at the other hand these effects increase wind stress curl and associated vertical transports. Getting more specific, we can now (based on our analysis of air-sea heat fluxes, kinetic energy supply, surface mixed layer dynamics, SST, surface currents and winds) pigeonhole ocean circulation model configurations into the following classes:

- **Coarse & no current/wind effect:** Eddy permitting Baltic Sea model configurations have only recently become computationally feasibly. Elder configurations do neither resolve eddies nor small-scale near-coastal circulation patterns. Additional simplifications in some of these configurations do comprise the neglect of current/wind effects in the wind-stress calculation. Insights gained in the present study suggest that these configurations (1) overestimate the supply of kinetic energy to the ocean, and (2) underestimate the horizontal inhomogeneity of the wind stress and its associated *Ekman Pumping*. As these spurious effects oppose one another the sign of the net effect on diapycnal transport is unclear. It is, however, evident that this class of configurations misses the substantial modulation of upwelling by the interaction of persistent near-coastal current features with the winds at major upwelling sites.

- **Coarse & current/wind effect:** Some of the elder coarse-resolution model configurations may include the representation of surface current/wind effects in their calculation of the wind stress. These configurations do not overestimate the supply of kinetic energy to the extent the latter class does. The overall *Ekman Pumping* is however – owed to the coarse resolution – still underestimated. Thus we speculate that the net effect on basin-scale is an underestimation of diapycnal fluxes. As concerns major upwelling sites, the modulating effects of persistent small-scale topographically-steered circulation (as described above) is not resolved.

- **Eddies & no current/wind effect:** Most contemporary model configurations strive to resolve the mesoscale. However, not all do explicitly account for current/wind effects. Among the reasons for the neglect are: (1) Increased computational performance (in cases where the calculation of the wind stress is performed by a coupler between the ocean and the atmospheric winds, these couplers can significantly increase wall-clock times). (2) Pragmatic avoidance of subtleties associated with the coupling: eddy-rich configurations are inherently non-linear. Taking current/wind effects into account adds another level level of non-linearity. The solutions become strongly dependent on the coupling time step and the choice is between small time steps requiring high computational costs and larger time steps resulting in diverging solutions. (3) The notion that the effect of surface currents are neglected because they are typically so much smaller than wind velocities.




The results in Section 3 suggest that these configurations overestimate the supply of energy and momentum supplied to the ocean. In turn, the energy available for diapycnal mixing is unrealistically high. This effect, however, is partly counterbalanced by spuriously reduced *Ekman Pumping*. As regards major upwelling sites, these configurations do resolve the near-coastal, persistent small-scale circulation. Its local effect on *Ekman Pumping* is, however, not considered.

–  **Eddies & current/wind effect:** the supply of kinetic energy and momentum to the ocean is not overestimated. Small-scale circulation patterns modulate the wind-induced up- and downwelling. The agreement with SST data is extraordinarily good (c.f. Fig. 8, 9 and 10 in Dietze et al., 2014, which show a comparison of *REF* with data).

## 5   Summary and Conclusion

In the present study, we compared two model simulations – one accounting for current/wind effects and one neglecting these.
We set out with theoretical considerations following Martin and Richards (2001) which proposed huge current/wind effects due to increased wind stress curl and associated *Ekmann Pumping*. Contrary to these theoretical consideration we find in our general ocean circulation model simulations that the major, prevailing effect of accounting for current/wind interaction is a large-scale reduction in the overall net supply of kinetic energy to the ocean. On basin scale, diabatic transport processes are thus, on average, rather reduced than enhanced. This reduction is reflected in a large-scale warming (cooling) of the ocean
surface in summer (winter).

Locally, however, the pattern is reversed and apparently in-line with our initial considerations: an analysis of winds and currents around the southernmost coast of Sweden (off Karlshamn and off the Kalmarsund) reveals a relatively persistent anti-cyclonic circulation. Consist with the argumentation of Martin and Richards (2001) we find increased *Ekmann Upwelling*, which is dominant and persistent enough to drive distinct local SST anomalies in summer. The climatological magnitude of
these anomalies is around 0.5 K and as such is prone to affect the formation of sea fog in those regions. Yet another effect of the upwelling of cold subsurface waters is associated with strong vertical gradients in nutrients such as phosphate and nitrate which are essential for autotrophic growth. During summer the sun-lit surface of the Baltic is typically depleted in these nutrients and all phytoplankton growth is impeded. Upwelling of cold subsurface waters which are enriched in nutrients do thus drive additional phytoplankton growth.

We conclude that surface current/wind effects are significant. Basin-scale effects correspond to $\approx 0.1$ K. Local, climatological effects may be reversed and peak at $\approx 0.5$ K. The timing in summer, when oligotrophic conditions prevail, in combination with the substantial magnitude suggests that local current/wind effects exert significant control on the complex biogeochemical cycling of the Baltic Sea.

*Acknowledgements.* We are grateful to Andreas Oschlies for providing essential long-term support. Integrations were performed on the
FB2/BM compute clusters *weil.geomar.de* and *wafa.geomar.de* located at the GEOMAR Helmholtz Centre for Ocean Research, Germany, Kiel, west-shore. We acknowledge help from Kai Grunau to set up, and maintain these clusters. We appreciate discussions with Robinson



Hordoir, SMHI, Sweden. This work is based on efforts from the Modular Ocean Model community (http://mom-ocean.org). Thank you for sharing your work so freely!



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
