# Peer review of "Effects of surface current/wind interaction in an eddy-rich general ocean circulation simulation of the Baltic Sea"

_Ocean Science, 2016_

## Referee Comment (RC1) · Anonymous Referee #1 · 16 May 2016

General

The author conduct two companion integrations of a regional model, one with the surface current included in the wind stress computation and another without. The main conclusion of the paper is that accounting for current-wind effects inhibits the total vertical exchange over the model domain, but enhances vertical exchange at major upwelling sites.

The overall approach and methods here seem generally sound. I find the narrative to be less than perfectly clear, however, mainly due to distracting phrases sprinkled throughout, a bit of a muddled notion of equivalence, and some clumsiness in crafting the storyline. The biggest problem I see is the notion that these results are inconsistent

with Martin and Richards (2001), who discussed Ekman pumping within coherent vortices, not the general impact of surface-current-wind interactions on vertical exchange. Here the authors seem to indicate that regions where vertical exchange is enhance with current-wind turned on are consistent with Martin and Richards (2001), even though these regions are not operating with the same physics; i.e., coastal upwelling south of Sweden versus coherent vortices.

Detailed comments

Introduction

pg 2, ln 9-10: the implication here is that adjoint methods are correcting stress estimates because of uncertainty in the stress formulation. Is that really the case? This methods is correcting for all source of uncertainty in the forcing, including the data describing the wind fields themselves.

pg 2, ln 21-22: I don't understand the sentence, "It is based on the success of the concept Ekman Pumping." Martin and Richards (2001) describe how eddy-wind interaction results in Ekman pumping within eddies.

Method

pg 3, ln 29: The word, "competitive" is not appropriate. It seems odd to report the resolution in nautical miles with the relevant metrics about scaling (Rossby radius) are in km. I would report the resolution in km.

pg 3, ln 32: KPP –> K-profile parameterization (KPP)

pg 4, ln 14: I find the phrase, "REF is identical to MOMBA 1.1" confusing. REF is a simulation and MOMBA 1.1 is a model? What "earlier Baltic Sea models" are you referring to?

pg 4, ln 23: I have difficulty parsing this text: "...detailed in Large and Yeager (2004); Large (2006) which has matured to a reference in the field (e.g. Griffies et al., 2014)."

[Figure]

What does that mean?

pg 4, ln 24: This sentence is unnecessarily complicated: "The setup noCW is identical to REF except for that the traditional (similar to, e.g., Meier et al., 1999, their Eq. 30), physically less plausible way to force an ocean model, which neglects the effect of surface currents on the wind stress, is applied." Also, use of the word "traditional" may be confusing to some readers with different levels of experience with ocean modeling— it is not necessary to characterize the approach this way.

pg 4, ln 30: What does "...to an apparently especially realistic model behaviour" mean? A period where the model compares especially favorably to observations?

pg 4, ln 31: why is bit-reproducibility relevant here?

Results

Fig 1. add panel showing SST and heat flux time-series?

pg 5, ln 10-17: I find this explanation hard to follow. What is the change in mean SST? Does stratification increase in REF relative to noCW? What happens to MLD? It seems that this is an obtuse angle from which to attack the differences in the simulations.

pg 5, ln 19: This sentence, "A gedankenexperiment reveals that by accounting for the ocean's movement in the calculation of wind stress exerted on the ocean's surface – overall – less energy is transferred to the ocean: winds and surface currents can – in addition to having a perpendicular component to one another – either oppose one another, or run along into the same direction" is confusing: why resort to a thought experiment when you have actual numerical experiments? What are you actually saying? Perhaps present Fig 2 first, then describe the mechanisms operating to cause this change.

pg 6, ln 1: Fig 2 confirms that there is less net energy transferred—if you rely on the reader to spatially integrate the difference field. Maybe point out that this is what you really mean.

pg 6, ln 5: Fig 3 looks like it has some mesoscale variability retained in the climatological field. Are the wintertime difference really the same sign everywhere? This figure indicates that mixed layers are not shoaling everywhere. This is not reflected in the text—again, you are leaving out a step, it is the spatial integral of this map, not the map itself, that indicates net shoaling over the domain.

pg 6, ln 9: "supply" –> "transfer"

pg 7, ln 9: I would have said these winds are southwesterly.

Fig 6: how is persistent defined? It's okay to provide a reference, but we should at least be provided with some minimum information to interpret what's plotted.

pg 7, ln 13: "Consistently, " –> "Consistent"

Discussion

pg 8, ln 5-7: Martin and Richards (2001) point to Ekman pumping in the interior of eddies. You only have an inconsistency, then, if you posit that the ocean surface is wholly dominated by coherent vortices. I am not sure that the Martin and Richards (2001) result can really be extended to make inferences about net momentum transfer with and without the surface-current effect on wind stress.

pg 10, ln 5: "...prevails [over] the effects...." It's not clear what is meant be "increase horizontal inhomogeneity." Please remind the reader of this concept as discussed in the introduction.

pg 11, ln 4: I don't think you have demonstrated this. You have presently only mean quantities, leaving open the possibility of effects with cancelation.

Summary and Conclusion

pg 12, ln 18: I don't see how this is consistent; you are talking about coastal upwelling and Martin and Richards (2001) discuss Ekman pumping within coherent vortices.

---

## Referee Comment (RC2) · Anonymous Referee #2 · 14 Jun 2016

Thi manuscript present a series of simulations that aim to quantify the influence of surface current effects on the surface stress and the resulting vertical exchange in the Baltic Sea. Overall, the is an interesting project and the manuscript is well written. I have one major comment regarding the experimental design and two minor comments:

Major: - The two simulations, one with and one without the relative wind correction, result in very different eddy fields. The simulation without the relative wind correction has much higher EKE. As a result of this fundamental difference in the two solution, I do not see a clear path as to how one could use this comparison to quantify the influence of the relative wind correction on vertical exchange. This has been one of the primary critaizum of previous efforts to make such comparisons (e.g., Eden and

Dietze, 2009). Without some sort of correction to account for the differences between the magnitude of the kinematic variability between the two simulation, the reader is left to wonder if the difference highlighted by the authors are indeed a result of the relative wind correction, or just the manifestation of a (likely) significantly less energetic solution in the simulations including the influence of the surface current on the surface stress. This could be address by redoing the simulations and using some other adjustment to bring the EKE of the two solutions closer together. Another option, and likely the easier one, would be to focus on mesoscale features and how the vertical exchange between them differer in the two simulations. This would also be more in-line with the current and previous research into this topic that highlights the influence of eddy-induced surface currents on imparting a curl in the surface stress.

Minor: - The discussion of how these results compare to some of the most important previous works in this field is missing. Once particularly appalling omission is the discussion of how this work builds on the fundamental work by Dewer and Flierl, 1987.

- On page 11, starting at line 4, the authors state that the inclusion of the relative wind correction on the surface stress "does not drive any additional near-surface diapycnal transport ..." This is not surprising as the use of the relative wind generates upwelling and downwelling, which alone, do not drive diapycnal transport. As such, this statement is moot.

---

## Author Comment (AC1) · 28 Jun 2016

Answer to Anonymous Referee 1 The referee's comments are typed in **bold**.

**I find the narrative to be less than perfectly clear, however, mainly due to distracting phrases sprinkled throughout, a bit of a muddled notion of equivalence, and some clumsiness in crafting the storyline.**

We acknowledge the referee's time and effort. We are especially thankful for his many constructive comments and suggestions!

[Figure]

**The biggest problem I see is the notion that these results are inconsistent with Martin and Richards (2001), who discussed Ekman pumping within coherent vortices, not the general impact of surface-current-wind interactions on vertical exchange. Here the authors seem to indicate that regions where vertical exchange is enhance with current-wind turned on are consistent with Martin and Richards (2001), even though these regions are not operating with the same physics; i.e., coastal upwelling south of Sweden versus coherent vortices.**

The main topic of the paper are "effects of surface current/wind interaction in an eddy-rich general ocean circulation simulation of the Baltic Sea". It is motivated by (1) published theoretical considerations which hint towards a strong effect of surface current/wind interaction, (2) by the fact that, only very recently, the spatial resolution of Baltic Sea model configurations allows for a fairly realistic representation of surface currents. Combining (1) and (2) raises the question if and to what extent previous (coarser resolution) model configurations are flawed - a question which is highly relevant in the Baltic as all projections into a warming future are based on (coarser resolution) models that do miss most or even all of the current/wind effect.

The referee is right in pointing out that we did not prove Martin and Richards (2001) wrong. The referee is also right in pointing out that coastal upwelling off Sweden is not to be confused with coherent vortices. We will revise the manuscript rephrasing all respective clumsy passages with the aim to make the following line of thought much clearer: (1) theoretical considerations suggest that surface current/wind interaction may give rise to substantial vertical upwelling and downwelling. (2) This raises the question if and to what extent previous (coarser resolution) model configurations are flawed in terms of their vertical transports of heat and nutrients. (3) To explore this we compare two simulations: one comprising the surface current/wind effect with one neglecting the effect. (4) We find that vertical exchange is, on average, damped.

Locally, however, as e.g. south of Sweden, the vertical exchange is increased.

**pg 2, In 9-10: the implication here is that adjoint methods are correcting stress estimates because of uncertainty in the stress formulation. Is that really the case? This methods is correcting for all source of uncertainty in the forcing, including the data describing the wind fields themselves.**

The implication is not the case. Rather: the wind stress formulation is so uncertain that modifications within is substantial uncertainty can "compensate" most of the other sources of uncertainty. We will clarify our reasoning in the revised version of the manuscript or delete the respective paragraph.

**pg 2, In 21-22: I don't understand the sentence, "It is based on the success of the concept Ekman Pumping." Martin and Richards (2001) describe how eddy-wind interaction results in Ekman pumping within eddies.**

We will delete " ... the success of ..." in the revised version of the manuscript.

**pg 3, In 29: The word, "competitive" is not appropriate.**

We will rephrase to " ... competitive compared to other Baltic Sea configurations ... ".

**It seems odd to report the resolution in nautical miles with the relevant metrics about scaling (Rossby radius) are in km. I would report the resolution in km.**

O.K.

**pg 3, ln 32: KPP –> K-profile parameterization (KPP)**

O.K.

**pg 4, ln 14: I find the phrase, "REF is identical to MOMBA 1.1" confusing. REF is a simulation and MOMBA 1.1 is a model? What "earlier Baltic Sea models" are you referring to?**

MOMBA 1.1 is a configuration of GFDL's Modular Ocean Model. We will clarify the issue in the revised version of the Manuscript. As concerns "earlier Baltic Sea models" we will add respective references - among them:

Meier, M. H. E., Döscher, R., Coward, A. C., Nycander, J., Doeoes, K. (1999). RCO-Rossby Centre regional ocean climate model. Model description (version 1.0) and first results from the hindcast period 1992/93. SMHI Reports. Oceanography (Sweden).

Meier, H. E., Döscher, R., Faxén, T. (2003). A multiprocessor coupled ice‐ocean model for the Baltic Sea: Application to salt inflow. Journal of Geophysical Research: Oceans, 108(C8).

Funkquist, L. and Kleine, E.: HIROMB - An introduction to HIROMB, an operational baroclinic model for the Baltic Sea, Tech. rep., SMHI, 2007.

**pg 4, ln 23: I have difficulty parsing this text: "...detailed in Large and Yeager (2004); Large (2006) which has matured to a reference in the field (e.g. Griffies et al., 2014)." What does that mean?**

We will delete " ... which has matured to a reference in the field ... " in the revised version of the manuscript.

**pg 4, ln 24: This sentence is unnecessarily complicated: "The setup noCW is identical to REF except for that the traditional (similar to, e.g., Meier et al., 1999, their Eq. 30), physically less plausible way to force an ocean model, which neglects the effect of surface currents on the wind stress, is applied." Also, use of the word "traditional" may be confusing to some readers with different levels of experience with ocean modeling— it is not necessary to characterize the approach this way.**

Agreed. We will rephrase in the revised version of the manuscript accordingly.

**pg 4, ln 30: What does "...to an apparently especially realistic model behaviour" mean? A period where the model compares especially favorably to observations?**

Yes. We will rephrase this clumsy sentence.

**pg 4, ln 31: why is bit-reproducibility relevant here?**

This is just for the sake of completeness. We will add some explanation here.

**Fig 1. add panel showing SST and heat flux time-series?**

Very good idea.

**pg 5, ln 10-17: I find this explanation hard to follow. What is the change in mean SST? Does stratification increase in REF relative to noCW? What happens to**

**MLD? It seems that this is an obtuse angle from which to attack the differences in the simulations.**

We will avoid the expression "thermal momentum" in the revised version of the manuscript. We believe that this inventing of new terminology made it hard to follow our argument. Here is what we were trying to say:

We find:
(1) A damped amplitude of the seasonal cycle of air-sea heat fluxes.
(2) An increased amplitude of the seasonal cycle in sea surface temperature.

We conclude that a reduced seasonal flux variability drives an enhanced surface temperature variability. Thus the water column subject to seasonal heating and cooling from the surface must be shallower.

We will clarify this in the revised version of the manuscript. We will also add the information that the basin-averaged MLD (defined by a bulk-Richardson number following Large et al .1994) gets shallower - in line with our reasoning.

**pg 5, ln 19: This sentence, "A gedankenexperiment reveals that by accounting for the ocean's movement in the calculation of wind stress exerted on the ocean's surface –overall – less energy is transferred to the ocean: winds and surface currents can – in addition to having a perpendicular component to one another – either oppose one another, or run along into the same direction" is confusing: why resort to a thought experiment when you have actual numerical experiments? What are you actually saying? Perhaps present Fig 2 first, then describe the mechanisms operating to cause this change.**

We will reformulate the corresponding sentence.

**pg 6, ln 1: Fig 2 confirms that there is less net energy transferred—if you rely on the reader to spatially integrate the difference field. Maybe point out that this is what you really mean.**

O.K.

**pg 6, ln 5: Fig 3 looks like it has some mesoscale variability retained in the climatological field. Are the wintertime difference really the same sign everywhere? This figure indicates that mixed layers are not shoaling everywhere. This is not reflected in the text—again, you are leaving out a step, it is the spatial integral of this map, not the map itself, that indicates net shoaling over the domain.**

We will clarify that we refer to spatial averages and we will add an explanation We regarding the "remaining mesoscale variability". The explanation will state that the respective variability is the result of differences in persistent current features which are correlated with topography.

**pg 6, ln 9: "supply" –> "transfer"**

O.K.

**pg 7, ln 9: I would have said these winds are southwesterly.**

We will change that to: " ... the winds' persistency decreases as they travel in a north eastward direction"

---

## Author Response (AR1)

Kiel, 11th July 2016

Dear Editor,

Please consider our revised manuscript "Effects of surface current/wind interaction in an eddy-rich general ocean circulation simulation of the Baltic Sea" for publication in Ocean Science.

Both reviewers were concerned that the presentation in the previous version of the manuscript implied that our results are inconsistent with findings of others regarding the eddy/wind effect. The reviewers were right that this could not be proved with our experimental setup. We are sorry for the confusion. This has not been our intention.

The aim of the manuscript is to explore the effects of surface current/wind interaction in an eddy-rich general ocean circulation of the Baltic Sea rather than proving Martin and Richards (2001) right or wrong. To make this clearer we rephrased respective sentences and paragraphs (as you will see in the point-by-point responses to the reviewers comments) and deleted "eddy/wind effect" in the abstract.

We acknowledge the time and effort of the two anonymous reviewers. We think they helped us to make the revised manuscript more appealing to a wider audience.

In any case, thank you for your time!

Yours sincerely,
the authors

Answer to Anonymous Referee #1
The referee's comments are typed in **bold**.

**I find the narrative to be less than perfectly clear, however, mainly due to distracting phrases sprinkled throughout, a bit of a muddled notion of equivalence, and some clumsiness in crafting the storyline.**

We acknowledge the referee's time and effort. We are especially thankful for his many constructive comments and suggestions!

**The biggest problem I see is the notion that these results are inconsistent with Martin and Richards (2001), who discussed Ekman pumping within coherent vortices, not the general impact of surface-current-wind interactions on vertical exchange. Here the authors seem to indicate that regions where vertical exchange is enhance with current-wind turned on are consistent with Martin and Richards (2001), even though these regions are not operating with the same physics; i.e., coastal upwelling south of Sweden versus coherent vortices.**

The main topic of the paper are "effects of surface current/wind interaction in an eddy-rich general ocean circulation simulation of the Baltic Sea". It is motivated by (1) published theoretical considerations which hint towards a strong effect of surface current/wind interaction, (2) by the fact that, only very recently, the spatial resolution of Baltic Sea model configurations allows for a fairly realistic representation of surface currents. Combining (1) and (2) raises the question if and to what extent previous (coarser resolution) model configurations are flawed - a question which is highly relevant in the Baltic as all projections into a warming future are based on (coarser resolution) models that do miss most or even all of the current/wind effect.

The referee is right in pointing out that we did not prove Martin and Richards (2001) wrong. The referee is also right in pointing out that coastal upwelling off Sweden is not to be confused with coherent vortices. We revised the manuscript rephrasing all respective clumsy passages with the aim to make the following line of thought much clearer: (1) theoretical considerations suggest that surface current/wind interaction may give rise to substantial vertical upwelling and downwelling. (2) This raises the question if and to what extent previous (coarser resolution) model configurations are flawed in terms of their vertical transports of heat and nutrients. (3) To explore this we compare two simulations: one comprising the surface current/wind effect with one neglecting the effect. (4) We find that vertical exchange is, on average, damped. Locally, however, as e.g. south of Sweden, the vertical exchange is increased.

Specifically we:
- deleted the reference to "eddy/wind" in the abstract (now on pg. 1, ln. 4)
- rephrased the summary and conclusions (now on pg. 13, ln. 5 to ln. 18)

**pg 2, ln 9-10: the implication here is that adjoint methods are correcting stress estimates because of uncertainty in the stress formulation. Is that really the case? This methods is correcting for all source of uncertainty in the forcing, including the data describing the wind fields themselves.**

The implication is not the case. We are sorry for the confusion. Rather: the wind stress formulation is so uncertain that modifications within is substantial uncertainty can "compensate" most of the other sources of uncertainty. We rephrased the respective sentence (now on pg. 2, ln. 10 to 11).

**pg 2, ln 21-22: I don't understand the sentence, "It is based on the success of the concept Ekman Pumping." Martin and Richards (2001) describe how eddy-wind interaction results in Ekman pumping within eddies.**

We will deleted " ... the success of ..." (now on pg. 2, ln. 21).

**pg 3, ln 29: The word, "competitive" is not appropriate.**

We removed the word (now on pg. 3 ln.29).

**It seems odd to report the resolution in nautical miles with the relevant metrics about scaling (Rossby radius) are in km. I would report the resolution in km.**

We report the resolution now in km (pg. 3 ln 30). We keep a reference to the nautical mile, though, because it has been "a computational barrier" in the Baltic Sea Modelling community for some time.

**pg 3, ln 32: KPP –> K-profile parameterization (KPP)**

Changed, now on pg. 4, ln. 1.

**pg 4, ln 14: I find the phrase, "REF is identical to MOMBA 1.1" confusing. REF is a simulation and MOMBA 1.1 is a model? What "earlier Baltic Sea models" are you referring to?**

We clarified/specified this; now on pg.4 ln. 13 to 15.

**pg 4, ln 23: I have difficulty parsing this text: "...detailed in Large and Yeager (2004); Large (2006) which has matured to a reference in the field (e.g. Griffies et al., 2014)." What does that mean?**

We deleted " ... which has matured to a reference in the field ... " (now on pg. 4, ln. 23 to ln. 24).

**pg 4, ln 24: This sentence is unnecessarily complicated: "The setup noCW is identical to REF except for that the traditional (similar to, e.g., Meier et al., 1999, their Eq. 30), physically less plausible way to force an ocean model, which neglects the effect of surface currents on the wind stress, is applied." Also, use of the word "traditional" may be confusing to some readers with different levels of experience with ocean modeling— it is not necessary to characterize the approach this way.**

We rephrased; now on pg. 4, ln. 25 to ln. 27.

**pg 4, ln 30: What does "...to an apparently especially realistic model behaviour" mean? A period where the model compares especially favorably to observations?**

Yes. Rephrased (now on pg. 4, ln. 31).

**pg 4, ln 31: why is bit-reproducibility relevant here?**

We added some information (now on pg. 4, ln. 32 to pg. 5 , ln. 2).

**Fig 1. add panel showing SST and heat flux time-series?**

Very good idea! We changed Fig. 1 accordingly (now on pg. 6).

**pg 5, ln 10-17: I find this explanation hard to follow. What is the change in mean SST? Does stratification increase in REF relative to noCW? What happens to MLD? It seems that this is an obtuse angle from which to attack the differences in the simulations.**

Sorry! We rephrased the section (now on pg. 5, ln. 1 to ln. 32) and modified Fig. 1 following your very good idea.

**pg 5, ln 19: This sentence, "A gedankenexperiment reveals that by accounting for the ocean's movement in the calculation of wind stress exerted on the ocean's surface –overall – less energy is transferred to the ocean: winds and surface currents can – in addition to having a perpendicular component to one another – either oppose one another, or run along into the same direction" is confusing: why resort to a thought experiment when you have actual numerical experiments? What are you actually saying? Perhaps present Fig 2 first, then describe the mechanisms operating to cause this change.**

We rephrased the whole section (now on pg. 5, ln. 1 to ln. 32) and hope it is much clearer now.

**pg 6, ln 1: Fig 2 confirms that there is less net energy transferred—if you rely on the reader to spatially integrate the difference field. Maybe point out that this is what you really mean.**

We added this information (now on pg. 6, ln. 1).

**pg 6, ln 5: Fig 3 looks like it has some mesoscale variability retained in the climatological field. Are the wintertime difference really the same sign everywhere? This figure indicates that mixed layers are not shoaling everywhere. This is not reflected in the text—again, you are leaving out a step, it is the spatial integral of this map, not the map itself, that indicates net shoaling over the domain.**

We clarified this (now on pg. 6 ln. 6 to 8).

**pg 6, ln 9: "supply" –> "transfer"**

Adopted; now on pg. 7, ln.2.

**pg 7, ln 9: I would have said these winds are southwesterly.**

Thank you! Now on pg. 8, ln. 1.

**Fig 6: how is persistent defined? It's okay to provide a reference, but we should at least be provided with some minimum information to interpret what's plotted.**

We added the definition (now on pg. 11, caption of Figure 7).

**pg 7, ln 13: "Consistently, " –> "Consistent"**

We made this mistake two times - thank you! Corrected; now on pg. 6, ln. 3 and pg. 8, ln. 5.

**pg 8, ln 5-7: Martin and Richards (2001) point to Ekman pumping in the interior of eddies. You only have an inconsistency, then, if you posit that the ocean surface is wholly dominated by coherent vortices. I am not sure that the Martin and Richards (2001) result can really be extended to make inferences about net momentum transfer with and without the surface-current effect on wind stress.**

We changed to "... apparently inconsistent with OUR initial considerations ..." (pg. 8, ln. 10).

Our initial considerations are based on ideas of Dewar and Flierl (1987) and Martin and Richards (2001). But the reviewer is right: reverse reasoning is not admissible here i.e. the fact that our initial considerations were not met in our Baltic Sea model does not prove neither Dewar and Flierl (1987) nor Martin and Richards (2001) wrong. We hope that this becomes much clearer now that we rephrased pg. 13, ln. 5 to ln. 14.

**pg 10, ln 5: "...prevails [over] the effects...." It's not clear what is meant be "increase horizontal inhomogeneity." Please remind the reader of this concept as discussed in the introduction.**

Rephrased, now pg. 9, ln.9 to 10.

**pg 11, ln 4: I don't think you have demonstrated this. You have presently only mean quantities, leaving open the possibility of effects with cancelation.**

True. We will change the sentence from " ... any additional near-surface ... " to " ... any additional net near-surface ...". Now on pg. 11, ln. 9.

**pg 12, ln 18: I don't see how this is consistent; you are talking about coastal upwelling and Martin and Richards (2001) discuss Ekman pumping within coherent vortices.**

We changed the corresponding sentence to "Consistent with the argumentation of D&F(1987) and M&R(2001) concerning the effects of surface currents on Ekman Upwelling these coastal anti-cyclonic surface currents effect an additional upwelling that is dominant and persistent enough to drive distinct local SST anomalies in summer."

Answer to Anonymous Referee #2
The referee's comments are typed in **bold**.

We acknowledge the refee's time and effort!

**The two simulations, one with and one without the relative wind correction, result in very different eddy fields. The simulation without the relative wind correction has much higher EKE. As a result of this fundamental difference in the two solution, I do not see a clear path as to how one could use this comparison to quantify the influence of the relative wind correction on vertical exchange. This has been one of the primary critaizum of previous efforts to make such comparisons (e.g., Eden and Dietze, 2009). Without some sort of correction to account for the differences between the magnitude of the kinematic variability between the two simulation, the reader is left to wonder if the difference highlighted by the authors are indeed a result of the relative wind correction, or just the manifestation of a (likely) significantly less energetic solution in the simulations including the influence of the surface current on the surface stress. This could be address by redoing the simulations and using some other adjustment to bring the EKE of the two solutions closer together. Another option, and likely the easier one, would be to focus on mesoscale features and how the vertical exchange between them differer in the two simulations. This would also be more in-line with the current and previous research into this topic that highlights the influence of eddy-induced surface currents on imparting a curl in the surface stress.**

Only very recently it has become computationally feasible to resolve small-scale surface currents such as coastal currents or part of the mesoscale variability in general ocean circulation model configurations of the Baltic Sea. Among other processes, this - for the first time - introduces a new mechanism: the surface current/wind effect. Theoretical considerations suggest that this should alter the vertical exchange of heat, salt and nutrients substantially. If so, previous models simulations of the Baltic which do not resolve this effect would be flawed. This is of some concern as such simulation are involved in political processes where expensive international decisions concerning future nutrient loads are made.

The aim of our manuscript "Effects of surface current/wind interaction in an eddy-rich general ocean circulation of the Baltic Sea" is to explore to what extend earlier model simulations/configurations of the Baltic Sea are potentially biased due to unresolved effects of surface current/wind interaction.

We deleted the reference to "eddy/wind" in the abstract (pg. 1, ln. 4) and stressed the above point on (pg. 13, ln. 5 to ln. 9).

**Minor: - The discussion of how these results compare to some of the most important previous works in this field is missing. Once particularly appalling omission is the discussion of how this work builds on the fundamental work by Dewer and Flierl, 1987.**

We will included the respective reference in the revised version (pg. 2, ln. 20).

**- On page 11, starting at line 4, the authors state that the inclusion of the relative wind correction on the surface stress "does not drive any additional near-surface diapycnal transport ..." This is not surprising as the use of the relative wind generates upwelling and downwelling, which alone, do not drive diapycnal transport. As such, this statement is moot.**

We rephrased the respective sentence (pg. 11, ln. 11 to 12) making clear that upwelling and downwelling - in combination with typical horizontal diffusive processes and air-sea buoyancy fluxes - do typically drive diapycnal fluxes: e.g. dense water is upwelled to the surface where it is heated by air sea fluxes. Thereby it looses density. When it is subsequently downwelled, the net effect is a diapycnal transport.

[revised manuscript text omitted]